

# Shallow landslides modeling using a particle finite element model with emphasis on landslide evolution

Liang Wang[1], Xue Zhang[2], and Stefano Tinti[1]

[1]Dipartimento di Fisica e Astronomia (DIFA), Settore di Geofisica, Università di Bologna, Viale Berti Pichat 8, 40127
[2]Department of Civil Engineering and Industrial Design, University of Liverpool, Liverpool, UK

**Correspondence:** Liang Wang (liang.wang2@unibo.it)

**Abstract.** Numerical modelling is a powerful tool to study the mechanism of landslides and constructs the foundation of many physically-based assessment methods applied to natural hazards. Usually, numerical analyses of landslides are carried out on the failure mechanism and on the propagation process separately. With the advantage of dealing with large deformation problems, the particle finite element method (PFEM), that is the particle extension of the traditional FEM, has the capability of simulating the entire evolution of the landslide from the generation to the deposition phase. To figure out the difference between a unified PFEM simulation and the usually adopted approaches that separate failure mechanism (static analysis) and run-out analysis (dynamic analysis), we implement a PFEM code that is applied first to a simple homogeneous slope model. Numerical results reveal that under the so-called critical condition the landslide comes to a stop with a slight modification of the original profile, while its profile is drastically changed if strength reduction is further applied. To test the capability of our model, we choose the 2013 Cà Mengoni landslide, northern Apennines, Italy, as a case study, since it behaved as if it were formed by homogeneous material. In virtue of the back-analysis of the run-out distance that we perform by using different material strength parameters, we show that the PFEM model is able to capture the variation of the observed landslide profile, and contributes to the understanding of the dynamics of the whole sliding process.

## 1 Introduction

A landslide is a mass movement that develops in time through several stages after the stability of a slope is disturbed. According to the movement type and its material components, Varnes built a classification system (Varnes, 1978) with 29 landslide types, which has been widely accepted and further developed in landslide studies (Hungr et al., 2014). Among these various mass movements, shallow landslides that are dominated by the deformation of soils or weathered rocks, significantly contributing to the evolution of landscapes, have received intensive attention from the perspective of the landslide risk assessment. The rainfall effects on pore water pressure condition and on material strength have been acknowledged as the main triggering factors of the generation of shallow landslides (Caine, 1980). Correspondingly, different empirical rainfall thresholds have been built for



the initiation of shallow landslides (Guzzetti et al., 2008). After destabilization, the failure mass moves from the source to the deposition area, which is also called the landslide propagation stage.

   Incorporating various gradually detailed experimental data and numerical models that analyze slope instability and mass-flow dynamics, different integrated models have been developed to assess landslide hazard (e.g., Miller and Sias, 1998; Be-
guería et al., 2009; Mergili et al., 2017). In fact, the whole process of landslide can be divided into the failure, post-failure and propagation stages by their distinct kinematic characteristics (Cascini et al., 2009). However, current models usually separate the failure mechanism from the propagation, due to the difficulties to couple the entire dynamic process under a unified computational framework. Therefore, most of the models only consider the landslide propagation that starts from rest assuming that that the slip surface is known, being determined by a previous slope stability analysis. The traditional Eulerian mesh-based
methods used to analyze slope stability are not adequate to accurately simulating large deformation problems. The propagation of landslides can be simulated by solving the so-called depth-averaged equations via numerical schemes that are derived from fluid mechanics and are able to account for the large deformation of the mass material occurring during landslide motion (e.g., Pastor et al., 2009; Pudasaini, 2012; Iverson and George, 2014). In addition to these, other methods have been proposed that solve the governing equations derived from solid mechanics, which appeals to the stability analysis of slope. To mention here
are the SPH method (Pastor et al., 2009; Bui et al., 2011), the MPM method (Andersen and Andersen, 2010; Abe et al., 2013), block-based methods (Tinti et al., 1997), interacting-particle methods (Gallotti and Tinti, 2019; Tinti and Gallotti, 2019) and PFEM (Oñate et al., 2011; Zhang et al., 2015). These methods are applied, or have the potential to be applied, to the entire land-slide evolution process starting from slope destabilization until mass deposition. For the simulation of the landslide dynamics, current studies mainly focus on validating the proposed numerical models by means of small scale simple-geometry examples
(Bui et al., 2011; Abe et al., 2013) without observation data (Andersen and Andersen, 2010). For the potential application of these methods, more investigations should be carried out especially focusing on the comparison of numerical results with available observation data.

   Among the mentioned methods, the particle finite element method (PFEM), which generates new mesh information at each time step, can be directly implemented into the traditional FEM with the help of domain detection (Cremonesi et al., 2011)
and variable mapping algorithms (Hu and Randolph, 1998). Based on the advantages of PFEM, the aim of this paper is to investigate the entire evolution of landslides, including the small scale slope and a real-world case. More specifically, first we analyze the dynamic evolution of simple-geometry landslides, which reveals that landslides can propagate significant distances if material strength values are much lower than the ones corresponding to the critical-condition determined by slope stability analysis. Then, to test the capability of PFEM of capturing landslide evolution, we study the case of the 2013 Cà Mengoni
landslide, since it represents a type of landslide that is widely observed in northern Apennines, Italy.

## 2   PFEM model configurations

As mentioned, PFEM is the particle extension of the classical FEM method. The FEM code we adopt here uses the mathematical programming technique to solve the optimization problem that is equivalent to the governing equations. Based on the



Hellinger–Reissner (HR) variational principle, the governing equations of incremental finite element analysis can be reformulated to the min-max optimization problem, which can be solved by freely available optimization solvers. The idea behind the process of searching the optimization solution is to obtain the solutions at the saddle point of the functional. This attractive approach was early used in the limit analysis of plastic problems (Krabbenhoft and Damkilde, 2003), elastoplastic finite element

analysis (Zhang et al., 2014) and recently has been applied to discrete element methods (Meng et al., 2018). The FEM code we use has been successfully applied to granular column collapses (Zhang et al., 2014), retrogressive failures of landslides in sensitive clays (Zhang et al., 2018) and submarine landslides (Zhang et al., 2019). More details on the mathematical programming and on the validation of the model can be found in the mentioned papers.

The PFEM method was first proposed to solve incompressible flows where large deformation and complicated contact
problems are involved. By means of the particle technique, the PFEM maintains all the classical advantages of the FEM for the evaluation of the integrals of the unknown functions and their derivatives are preserved, including the easiness to impose the boundary conditions and the use of symmetric Galerkin approximations (Idelsohn et al., 2004). The key feature of PFEM is that mesh nodes are viewed as particles that can move and even detach from the domain to which they originally belong. To do this, the modified 'alpha-shape' method Cremonesi et al. (2011) is employed to identify the computational domain. The recognized
domain is used to generate new mesh information, and further another step of variable mapping Hu and Randolph (1998) is adopted to map variables from old elements to new elements. Instead of developing new computational frameworks of other meshless methods, the particle finite element modeling technique can be straightforwardly extended within the existing FEM codes. Fig.1 is used here to explain the technique implemented in our PFEM model.

So far, the applications of particle methods in the entire evolution of landslides are quite few, especially in investigating
the slope model of real cases. The applications of PFEM to the "landslide dynamic evolution" were first conducted from the perspective of fluid mechanics, where the Bingham constitutive model was used to analyze the dam stability (Larese et al., 2013). Further, the PFEM was applied to the propagation of real landslide cases (Zhang et al., 2015; Cremonesi et al., 2017). Depending on the advantage that it is built on the solid FEM foundation and is able to deal with large deformation problems, the use of PFEM in landslides research should be further developed with more efforts.

## 3   Homogeneous soil slope

Before applying our PFEM model to a real-world landslide, we restrict our interest to the failure mechanism of a small-scale slope, which is usually chosen as a benchmark in slope stability analysis (Griffiths and Lane, 1999). The case we treat here was also analyzed through two different SPH codes by Bui et al. (2011) and by Peng et al. (2015), who both made use of the Drucker-Prager elastoplastic model with non-associated flow rule. The stability results obtained via SPH agree well with the
classical FEM analysis and the dynamic evolution is that the failure mass moves as a sliding block along the identified slip surface. In both SPH codes, the simulations were conducted with material parameters modified by prescribed reduction factors that were set close to the critical safety factor, which is the so-called critical failure condition for the slope. In contrast to the





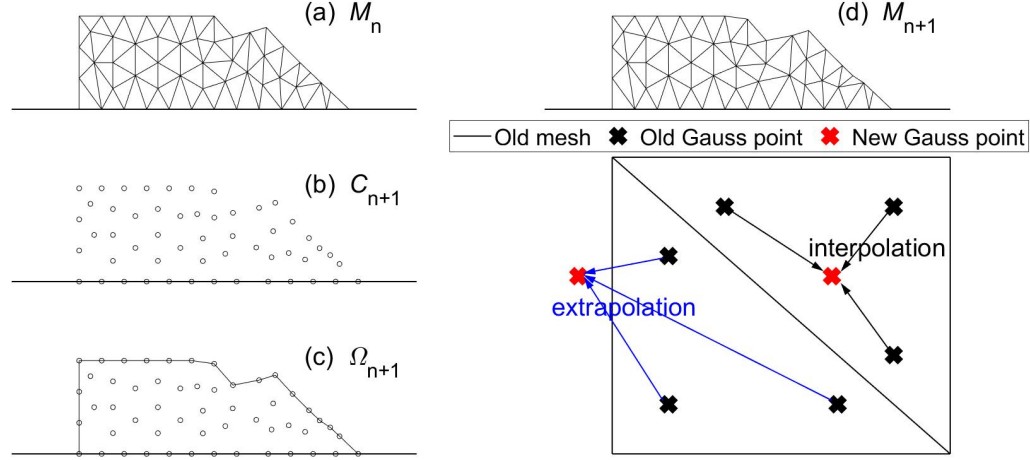

**Figure 1.** Steps for PFEM techniques in the time interval $[t_n, t_{n+1}]$. (1) Re-mesh step, from (a) to (d): The position information of mesh nodes at the instant $t_{n+1}$, denoted as $C_{n+1}$, is updated by exploiting the velocity information on the nodes of mesh $M_n$ at $t_n$. With the coordinates of $C_{n+1}$, the 'alpha-shape' method is used to identify the boundary nodes, which are used to generate mesh $M_{n+1}$. (2) Variable mapping: For the new node that stores variables, the corresponding old element that includes or is the closest to the new node (in the sense that its center has the shortest distance to the node) should be determined. Further, the values of variables at the new node can be interpolated or extrapolated with the shape functions and the values at the old nodes inside the element. In our case, the mixed triangular element is used. For more details one can refer to (Zhang et al., 2017).

failure mechanism depicted by Bui et al. (2011), the results by Peng et al. (2015) depict that the landslide stops with shallow profile modification, a feature that can be also observed in the shallow movements of landslides.

The two homogeneous slope models have the same simplified geometry (see Fig. 2), though with different lengths and material parameters (see Tab. 1). The investigated cases by Bui et al. (2011) and by Peng et al. (2015) are denoted as Case 1 and Case 2 respectively in this paper. It should be noted that the chosen dilation angle for Case 1 is the same as the one in the original example given by Griffiths and Lane (1999). In our numerical code we implement the non-associated Mohr-Coulomb constitutive model (Krabbenhoft et al., 2012), whose accuracy has been recently validated (Wang et al., 2019). The material parameters for soil including elastic modulus $E$, internal friction angle $\phi$, cohesion $c$, dilation angle $\psi$, Poisson's ratio $\nu$ and density $\rho$ can be found in Table 1.

## 3.1 Critical condition

The critical condition of a slope may be defined as the moment that the slope transits from a state of stability to instability. By means of the shear strength reduction method (SSR), the strength parameters, i.e. cohesion and internal friction angle, are





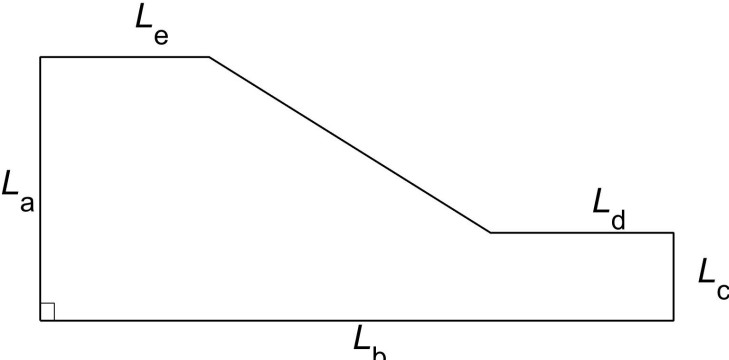

**Figure 2.** Geometry of the homogeneous slope model. Case1: $L_a$ =15m, $L_b$ =45m, $L_c$ =5m, $L_d$ =13m, $L_e$ =12m; Case2: $L_a$ =7m, $L_b$ =20m, $L_c$ =2m, $L_d$ =5m, $L_e$ =5m.

**Table 1.** Material parameters for homogeneous soil slope.

|  | $E\,(MPa)$ | $\phi\,(°)$ | $c\,(kPa)$ | $\psi\,(°)$ | $\nu$ | $\rho\,(kg/m^3)$ |
|---|---|---|---|---|---|---|
| Case1 | 100 | 20 | 10 | 0 | 0.3 | 2000 |
| Case2 | 30 | 30 | 5 | 0 | 0.2 | 1850 |

The given values are elastic modulus $E$, internal friction angle $\phi$, cohesion $c$, dilation angle $\psi$, Poisson's ration $\nu$ and density $\rho$.

modified by the reduction factor ($RF$), the critical value for RF being approximately equal to the stability factor of the slope ($FOS$). The safety factors computed by static analysis in our model for Case 1 and Case 2 are 1.38 and 1.953 respectively and are comparable with the published values of 1.40 and 1.94 (Griffiths and Lane, 1999; Peng et al., 2015). In this section, we adopt the critical reduction factors $RF = 1.40$ and $RF = 1.953$ for the dynamic simulations to investigate how the slope

5   evolves under the critical condition. In the simulations, we chose the mesh element size with typical area of 0.25 $m^2$ for Case 1 and of 0.16 $m^2$ for Case 2, resulting in the corresponding total number of triangles of 2759 and of 1566. The adopted boundary conditions are that lateral nodes cannot have horizontal displacements and bottom nodes cannot move at all. The time step $\Delta t$ is set to 0.5 s, and the mesh is regenerated at each time step keeping the same typical element size. The final deposit profile we obtain for the two slopes with our PFEM model are displayed in Fig. 3.

10     From Fig.3 it can be seen that the PFEM model, which has solid foundations in continuum mechanics, describes that the homogeneous slope stops with slight modification of the original profile. This result also suggests that the weakening process during the post-failure stage influences the mobility of shallow landslide and debris flows, which will be proven in the next section.



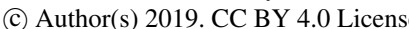



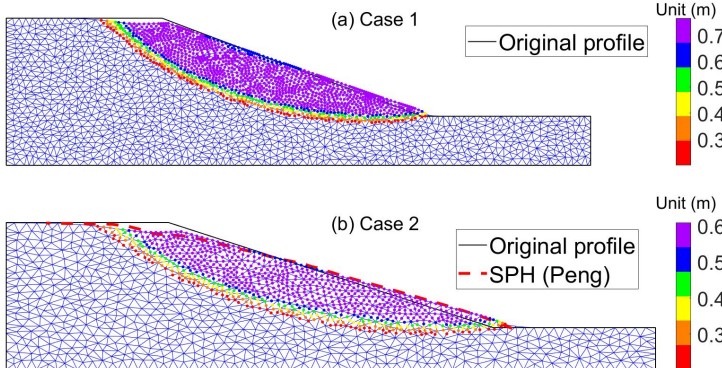

**Figure 3.** Final profiles of the slope with distribution of displacement for Case1 and Case2. The displacement field are shown for nodes that have a displacement over 0.2 m. PFEM simulations capture that the slopes deposit with little modification of the initial profile. The simulation for Case 2 agrees well with the SPH simulation (Peng et al., 2015)

## 3.2 Sensitive analysis of parameters

It is of interest to investigate the dynamic evolution of the slope when the material strengths are further reduced. Since Case 1 and Case 2 are based on the same slope geometry and the numerical results describe the same phenomenon, we can only analyze one of the two, such as for instance Case 2. By the same SSR technique, individual reduction factors $RFC$ and $RFF$ are implemented to reduce cohesion and internal friction angle respectively. To quantify the run-out distance of the sliding body, the average displacement $D$ is calculated by averaging the displacement of nodes that run distances more than 0.1 m. The value of $D$ computed by the critical condition is denoted as $D_c$. The influence of $RFC$ and $RFF$ on the ratio between $D$ and $D_c$ is plotted in Fig. 4. As shown in Fig.4, the reduction of cohesion $c$ and friction angle $\phi$ significantly contributes to the mobility of landslides. The weakening processes of $c$ and $\phi$ are attributed to the break of shear band and the rearrangement of grains (Iverson, 1997). The clearly sharp slope of the curve of $RFF$ indicates that the reduction of friction angle has larger influence on the mobility of landslides. The dynamics of the landslide is also shown through the evolution of the average velocity in Fig. 5. After failure ($t = 0s$), the main body achieves its peak velocity at $t = 1.5s$ and then decelerates to still within 4 s in total.

The final profiles with $RFC = 8$ and $RFF = 3$ are displayed on Fig. 6. It can be seen that the reduction of internal friction angle induces a deeper failure surface.

It is clear that the slope moves along the slip surface, which can also be identified by the equivalent plastic strain ($PEEQ$). If the cohesion is extremely low or the material is softened by the accumulated strains, the landslide can move with several slip surfaces. A good example is given in Zhang et al. (2017), in which a sub-landslide can be observed inside the sensitive clay landslide. The plastic strain zone along the surface is also observed in Fig. 7.





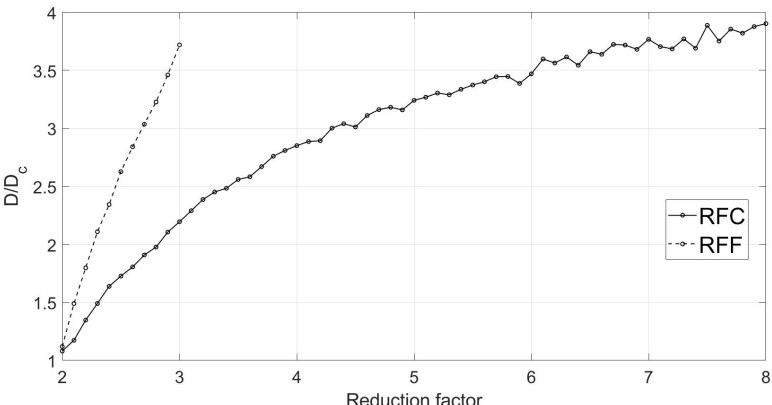

**Figure 4.** Relationship between the ratio $D/D_c$ and the reduction factors for cohesion and internal friction angle. The reduction factors are not further reduced beyond the limit shown in the graph, since the failure mass reaches the right lateral boundary of the numerical slope domain.

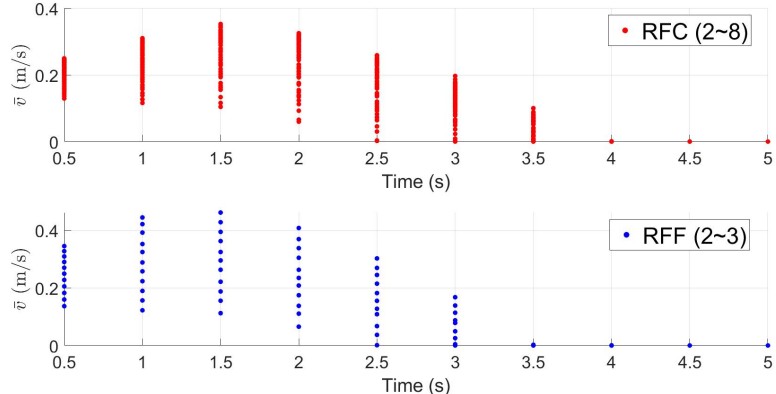

**Figure 5.** Evolution of the average velocity $\bar{v}$ of the sliding body, computed by averaging over the nodes that have the magnitude of velocity larger than $10^{-4}\ m/s$.

## 4 The 2013 Cà Mengoni landslide

To prove that our model can be implemented in geoscientific procedures to assess landslide risk, we choose the Cà Mengoni landslide, that occurred on 6 April 2013 in the northern Apennines, close to Castel dell'Alpi, a village in the province of Bologna, Italy, and we use it as a case study to investigate the capability of our model in capturing the final profile. The static stability analysis of the 2013 Cà Mengoni landslide was conducted by Berti et al. (2017), where the material strength parameters were back-analyzed in details. The landslide is a flysch rock slide, and is of a type quite common in northern Apennines. Previous studies (Berti et al., 2017; Ronchetti et al., 2009) indicated that, despite its complexity, the flysch behaves





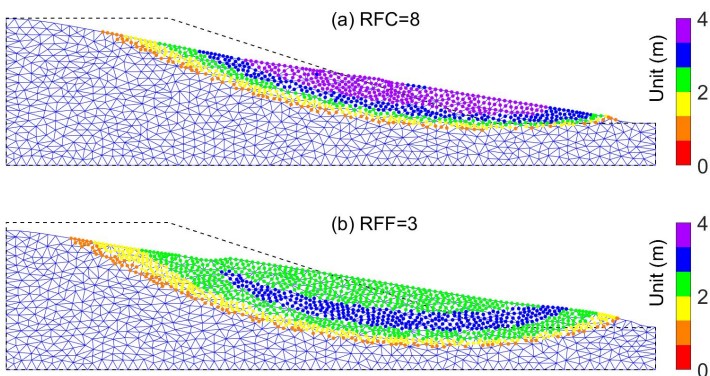

**Figure 6.** Final displacement profiles of the landslide with $RFC = 8$ and $RFF = 3$. Nodes are plotted with displacement over 1 m. Dash black line is the original profile.

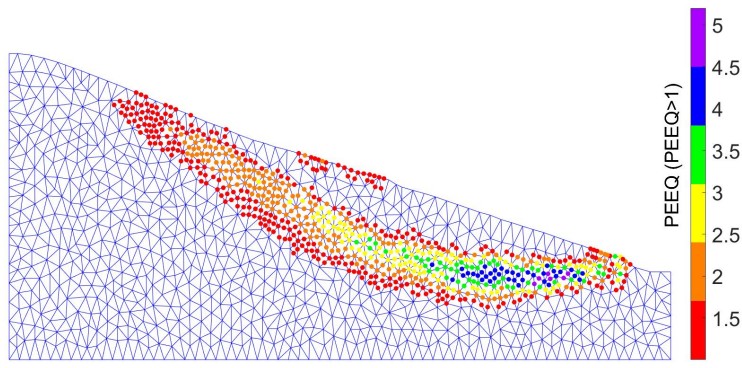

**Figure 7.** Contour of equivalent plastic strain ($PEEQ$) in the final profile of the slope with $RFC = 8$.

as a homogeneous medium at the slope scale. Additionally, we note that the final profile of the 2013 Cà Mengoni landslide is similar to the one of shallow landslides. Therefore, we take this as a hint that our model based on continuum mechanics can be implemented to investigate this historical event. The strategy is to use the PFEM model to investigate the failure mechanism and profile variation from the perspective of analyzing the final deposit with different parameters.

## 4.1 Background and slope model

The 2013 6th April Cà Mengoni landslide, located in the Apennines south of Bologna was triggered after a long period of rainfall with about 310 $mm$ in 30 days, and finally deposited on 8th, April. The main motion of the landslide occurred in the early afternoon of 6th, April and then slowed down reaching a velocity less than 1 $m/h$ in the late afternoon. The landslide,

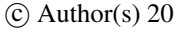



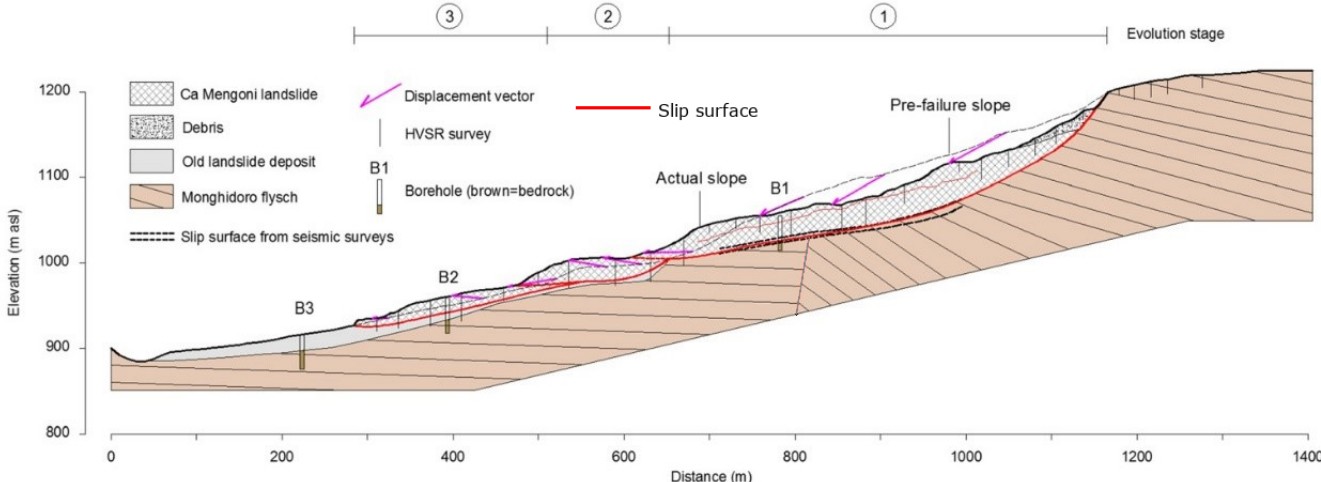

**Figure 8.** Evolution of the main cross-section of the Cà Mengoni landslide. Circled numbers refer to the stages of movement (after Berti et al. (2017)).

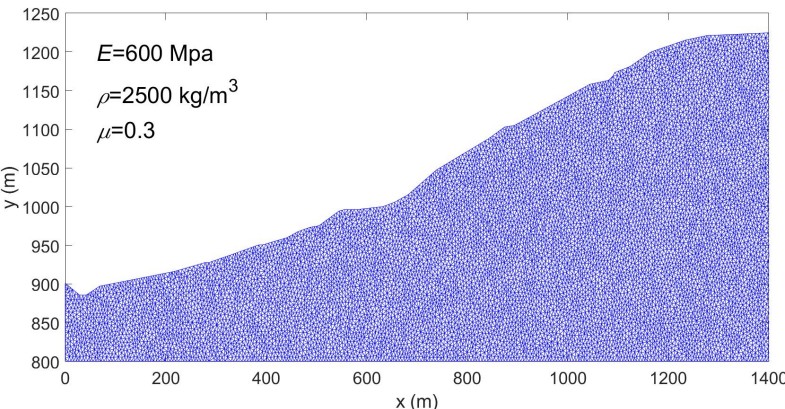

**Figure 9.** Numerical set-up for the slope model. The horizontal displacements vanish on the lateral boundaries. The bottom of the model is fixed. Elastic modulus $E$, density $\rho$ and Poisson's ratio $\nu$ are taken from Berti et al. (2017).

that has an estimated volume of 3 million $m^3$, was witnessed moving with an approximately peak velocity of 10 $m/h$. The landslide mainly consists of a cretaceous flysch that is made of a close alternation of sandstones, siltstones, and marls. The detailed field survey reveals that the original slope was initialized within the flysch substratum. More geological details can be found in Berti et al. (2017).





The original, intermediate and final profiles of the landslide are shown in Fig. 8. The corresponding 2D numerical mesh in Fig.9 consists of 21755 triangles with area of 25 $m^2$. Values for elastic modulus $E$, density $\rho$ and Poisson's ratio $\nu$ are taken from Berti et al. (2017) as well as the values for cohesion $c$ and internal friction angle $\psi$, that are 30 $kPa$ and 29° respectively. In the present model the material is one phase only, so the numerical slope model is stable with the chosen $c$ and $\phi$. Different reduction factors $RFC$ and $RFF$ are used to investigate the dynamic evolution of the landslide.

## 4.2  Numerical investigation

The dynamic evolution of landslides involves various mechanisms during different stages, while here we simulate the dynamic evolution of homogeneous material controlled by the gravity force and local topography. In this section, we concentrate on the mass flow movements within shallow depth by implementing different material strength parameters. In contrast to the conventional landslide models, where the materials are released without velocity, this present PFEM model provides more information on the identification of the potential failure zone.

### 4.2.1  Slip surface by static analysis

The associated and nonassociated Mohr-Columb model is implemented in our PFEM code to identify the slip surface of the slope. From the adopted values of cohesion and internal friction angle, the obtained $FOS$ are 1.5 and 1.545 respectively. The identified slip zones are shown in the $PEEQ$ plots in Figure 10. For the associated model, the toe of the slope starts to move when the reduction factor equals to 1.5, and this indicates a local failure of the landslide. Since the plastic zone depicted in Figure 10(a) does not show connection between the toe and the slope surface, a further check was carried out to check the robustness of FOS searching algorithm (Wang et al., 2019). With the associated Mohr-Coloumb model, the numerical landslide fails when $RF$ is higher than 1.5. This slip zone shown in Figure 10(b) is deeper than the one observed and the one obtained by the finite difference analysis by Berti et al. (2017), where the water table was set at the ground. The $PEEQ$ accumulates at the toe of the generated slip zone downhill and gradually decreases moving uphill. It should be noted that the slip surface obtained by means of the static analysis here is not the slip surface obtained by means of the dynamic analysis, which will be explained later on.

### 4.2.2  Dynamic evolution

The dynamic evolution of landslides involves several dynamic processes, such as the dynamic behavior of material and the redistribution of landslide slips. Unlike the single slip surface shown in Figure 10, simulations here present the generation of multi-slip surfaces during post-failure stages. The previous study on a simple geometry slope showed that the reduction of internal friction angle $\phi$ has a significant influence on the slip surface (see Fig. 6), contributing also to the generation of multiple slip surfaces.

The critical condition of $RF = 1.545$ identifies the stage where the plastic strain zone allows the slipping over an internal surface, which marks the landslide initiation. With the same strategy applied in the previous section for Case 2, we test different





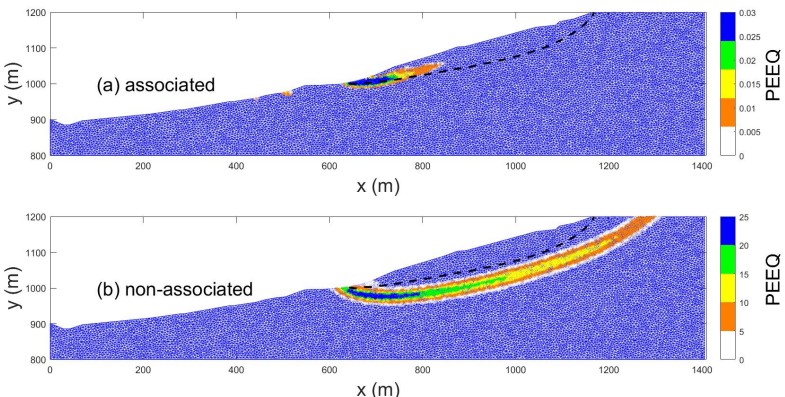

**Figure 10.** Contour of $PEEQ$ under the critical condition for the slope models that adopt associated and non-associated Mohr-Coulomb models (only nodes with $PFEQ$ value higher than $10\%$ of the maximum $PEEQ$ are plotted). Black dashed line represents the observed slip surface during the first stage of the landslide, which is shown in Figure 8.

values for $RFC$ and $RFF$. If not mentioned explicitly, the end flag of simulation is the minimum average velocity lower than $10^{-4}\ m/s$ and the time step is $\Delta t = 1s$. When the failure mass is still moving, a clear velocity distinction between the mass and the stable body can be seen. We checked all the cases that the failure mass is stopped with the specific velocity threshold, even by adopting a very large time step.

The final profile of the landslide using the critical reduced parameters gives the local failure of the initial slope displayed in Figure 11(a). Increasing $RFF$ makes the plastic zone closer to the observation data, as shown in Figures 11(b) and 11(d). Two cross slip surfaces are observed in Figure 11(d) and this can represent two different stages of the landslide. Figure 11(c) shows that the increment of cohesion does not change the local failure portrayed in Figure 11(a). It should be mentioned that the non-normalized $PEEQ$ values for the case of Figure 11(c) are generally higher than the ones of Figure 11(a), and that also

the $PEEQ$ values inside the sliding body are distributed more widely than in Figure 11(a). Interestingly, Figure 11(b) depicts a slip surface that agrees well with the slip surface of the first stage of the landslide. However, it cannot describe the whole stages of the landslide, and thus the deposit profile is not well captured.

      Simulations with various $RFF$ and $RFC$ values are also conducted to investigate the dynamic mechanism of landslide evolution, and the maximum displacement of the nodes is chosen as an evaluation index. As shown in Fig. 13, the landslide

can have a maximum displacement over 200 meters when $RFF$ is larger than 1.5. The slope of the isolines confirms that the landslide mobility is influenced much more by $RFF$ than by $RFC$.

### 4.2.3   Landslide deposit profile

With the observed data and the numerical results, the misfit index (Zaniboni and Tinti, 2014) can be calculated by means of operation on polygons. The polygons corresponding to the observed landslide deposit profile and to the numerical deposit



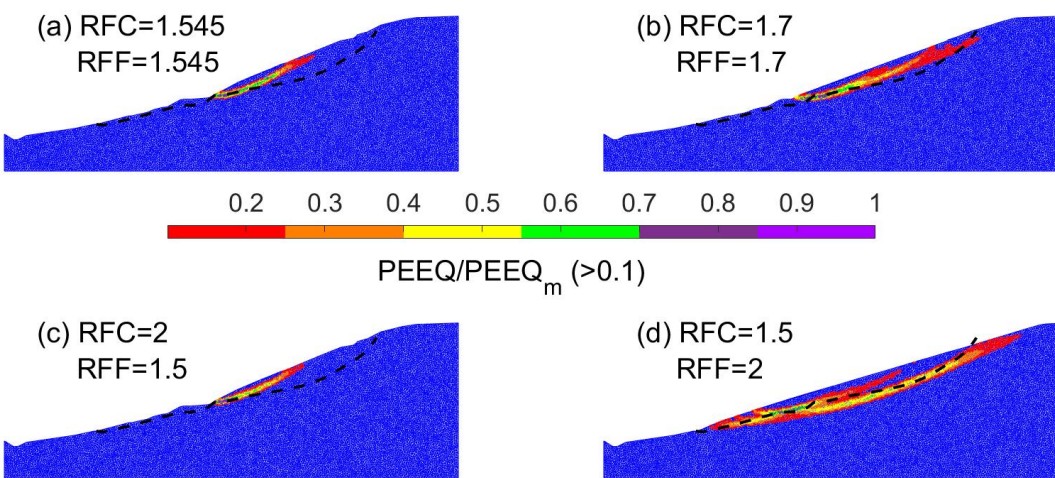

**Figure 11.** Normalized $PEEQ$ contour with final profile of landslide for four cases. The displayed $PEEQ$ is normalized to the peak value of $PEEQ$, denoted as $PEEQ_m$. Black dashed line is the observed slip surface.

profile are denoted as $P_O$ and $P_N$. Using $P_I$ for the polygon of the initial profile, the misfit index, designated by $MI$, is computed through the formula:

$$\Delta P_O = P_O \cup P_I - P_O \cap P_I \qquad (1)$$

$$\Delta P_N = P_N \cup P_I - P_N \cap P_I \qquad (2)$$

$$5 \quad MI = 1 - \frac{\Delta P_O \cap \Delta P_N}{\Delta P_O \cup \Delta P_N} \qquad (3)$$

According to Figure 14, the most similar numerical deposit profile might appear when the MI takes the smallest value. Considering that the values are quite close, we choose the four smallest values of $MI$ and plot the corresponding results in Figure 15. The run-out distance is well reproduced in the case (a), (c) and (d), while the case (b) indicates that with the value $RFF = 1.9$ one might obtain a slip surface more in agreement with the observation data.

10   The present PFEM simulation not only provides a reasonable deposit compared with the observed deposit profile, but also agrees quite well with the observed slip surface. This remarkable result indicates that: (1) the classical slope stability analysis might overestimate the strength parameters when it is applied to real landslide cases; (2) weakening process is quite important during the propagation of a landslide. Combining Figure 11(b), which captures the slip surface of the first stage and Figure 15, one may conclude that the observed slip surfaces and the run-out distance might be reproduced reasonably well by weakening

15   the material parameters.



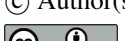

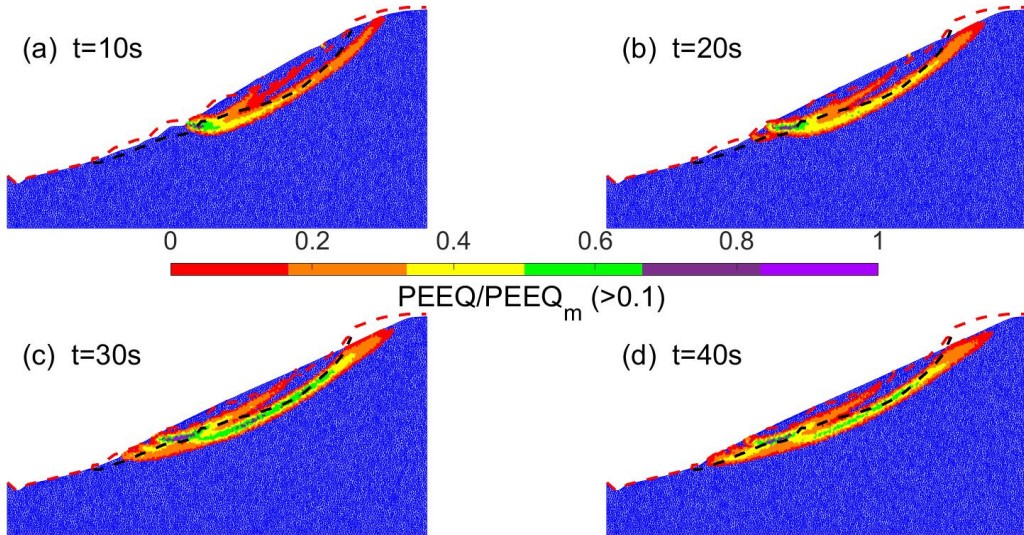

**Figure 12.** Snapshots of $PEEQ$ with $RFC = 1.5$, $RFF = 2$. It can be seen that two main slip surfaces form at the beginning of the landslide evolution. With different sliding velocities, different slide bodies are observed in (b), (c) and (d). Red and black dashed lines represent the actual slope and observed slip surface. The landslide is nearly stopped with $t = 40s$.

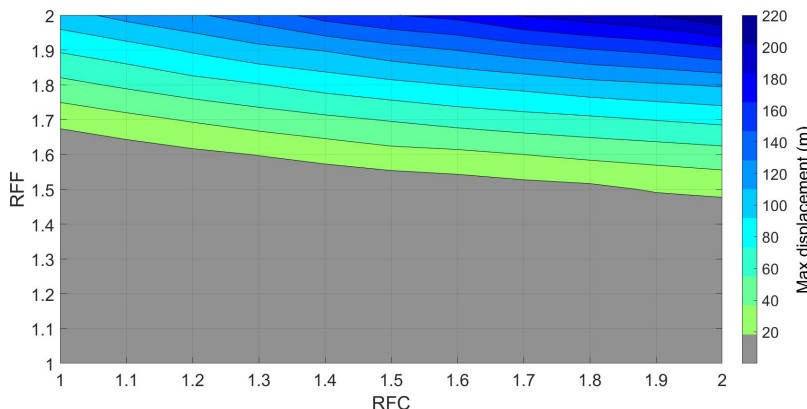

**Figure 13.** Relationship between the maximum displacement and the reduction factors $RFC$ and $RFF$. The simulations are conducted with $RFC$ and $RFF$ varying from 1 to 2 with equal step of 0.1.





**Figure 14.** Relationship between the $MI$ and the reduction factors $RFC$ and $RFF$ that are varied from 1 to 2 with equal interval of 0.1.

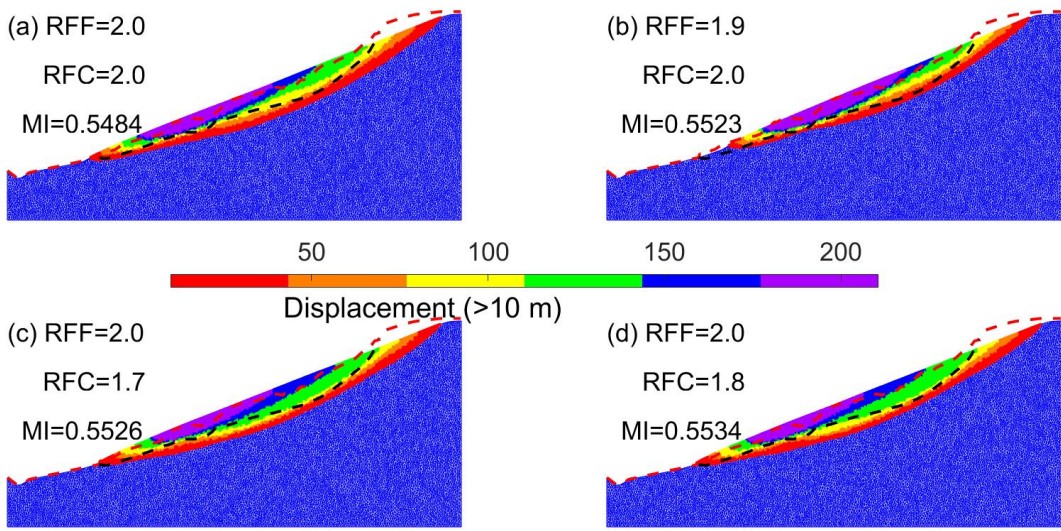

**Figure 15.** Final deposit profile with four cases. Red and black dashed lines represent actual slope and observed slip surface.

### 4.2.4 Weakening process

The previous simulations are carried out with the sudden reduction of strength parameters, where the reduction factors $RFF$ and $RFC$ are employed at the first dynamic time step (after the gravity balance calculations). In this section, a simple gradually increasing reduction factor law is implemented into the code to cast a light on the weakening process. In a first stage with a

5    duration $T_1 \sim 280s$ $RFF$ and $RFC$ are gradually increased from 1.5 until $RFF = 1.90$ and $RFC = 1.98$. After that, the





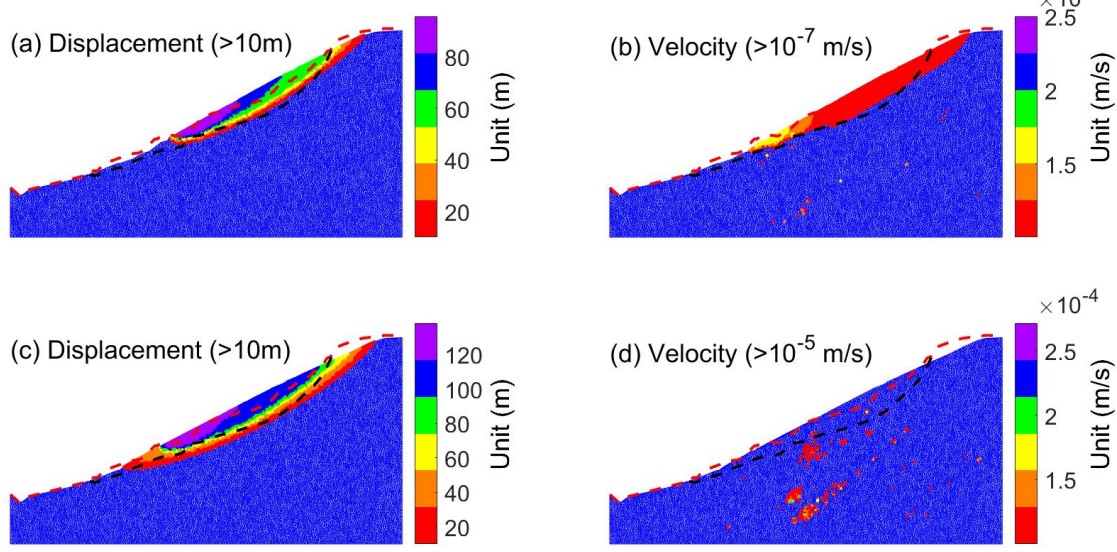

**Figure 16.** Final deposit profile for the weakening process. Red and black dashed lines represent actual slope and observed slip surfaces. The end flag for (a) and (b) is set when all node velocities fall lower than $10^{-7} m/s$. Further, the simulation continues with another weakening process of $RFF$. The simulation finally ends for (c) and (d) when nodes in the slip zone have negligible velocities, as shown in (d).

simulation remains with $RFF = 1.9$ and $RFC = 2$ until the general maximum velocity smaller than $10^{-5} m/s$ ($T_2 \sim 2900s$). The results for the first weakening process are shown in Figure 16(a) and 16(b). It can be seen that this quasi-static weakening process produces a result that is slightly different than the one shown in Figure 15(b). In Figure 16(a), the landslide has maximum displacement around 80 m at the front, which is smaller than in Figure 15(b). Further, we note that the sliding body

moves with extremely low velocity along the slip surface as shown in Figure 16(b).

Then, in another phase of duration, $T_3 \sim 1100s$, the factors are increased to the final values $RFF = 2.0$ and $RFC = 2.0$ and the consequence is that the landslide moves at a higher velocity before reaching the final deposit profile that is a good approximation of the observed profile (see Figs. 16(c) and 16(d)). Notice that the time intervals $T_1, T_2$ and $T_3$ have been selected with no reference to the observed time phases of the real Cà Mengoni landslide, but only with the purpose to investigate how

our PFEM model performs assuming time-dependent material parameters. This experiment suggests that the evolution time of the landslide is strongly governed by the weakening process of the sliding material, and that, when it is extremely slow, like in creeping, it can be interpreted as the progressive passage through varying conditions of quasi-equilibrium. In other words, it can be seen as a quasi-static process rather than as a full dynamic process. It also suggests that more physically-based weakening models, i.e. strain-accumulated model, should be further employed to investigate further this topic, as it deserves.

The present PFEM model, that describes the dynamic deformation of a homogeneous slope that is controlled by gravity, local topography and material strength, gives results that agree well with the observed deposit profile and slip surface of the



landslide when choosing appropriate reduction of internal friction angle $\phi$. This finding can contribute to the risk assessment of shallow landslides, where landslide movements are controlled by deformations. We note that with the current model we have not been able to predict exactly the dynamic evolution of the Cà Mengoni landslide, as we have obtained a landslide moving much faster than it was. This is not critical, however, since slow creeping can be obtained by applying proper weakening

laws. Furthermore, we point out that our model can provide more complete information on a landslide process, including the identification of the failure and influence areas, which is quite important for planning adequate mitigation strategies.

## 5    Conclusions

The accurate modeling of the entire process of landslide under a unified computational framework is a challenge for landslide studies and the significant developments of novel numerical approaches cast a light on solving this issue. To investigate and

explore the capability of meshless approaches, the PFEM approach was employed in this paper. The present PFEM model, which combines the conventional finite element analysis and the particle-based technique, can simulate the dynamic evolution of landslides, mainly focusing on the shallow deformation of a slope.

Based on typical examples in slope stability analysis (Griffiths and Lane, 1999), we observe that under critical conditions the failure mass deposits with slight modification of the original profile. This result indicates that there should be a further

weakening process that is effective in controlling landslide propagation. Further investigations show that the reduction of internal friction angle significantly contributes to the mobility of landslides.

Applying the PFEM model to the 2013 Cà Mengoni landslide, that is a typical mass-movement event in northern Appennines, Italy, and was controlled by the homogeneous deformation of weak rocks, we found that the multi-slip surface mechanism can be accounted for by our model. Indeed, the numerical results show that the slip-surface obtained by the dynamic analysis is

different from the one obtained by static analysis, since the plastic strain zones further develops during the movement of the failure mass, with a redistribution of stresses. Moreover, the model can provide a deposit profile and slip surfaces that well approximate the observed data if suitable material parameters are specified. The dynamic analysis, however, is not able to reproduce the observed duration of the process, which numerically occurs in the order of tens of seconds rather than in several hours. We have shown that the overall landslide duration can be modified by assuming simple weakening laws for the material

parameters. And that the final landslide picture is more sensitive to the final values taken by the material parameters than to the weakening time interval. In this paper, we have considered a maximum weakening time of about 4280 s instead of the observed time that was about two-day long (Berti et al., 2017), because of running time limitations. Our results show that more physically-based weakening models should be built, especially considering the effects of rainfalls on material properties, and also that technically the use of larger time step to reduce the computational time without affecting the physical significance of

the results should be further studied.

Certainly, the dynamic evolution of landslides is a complex process, and the present model reveals to be probably too simple to capture such a complicated dynamic mechanism in all cases. It performs well when the shallow movement of some slopes can be simplified as the motion of a homogenous material, which fits the assumptions of the present model that is based on



continuum mechanics. Future developments of our research are to implement more advanced constitutive models accounting for material weakening and rainfall effects. Also, the reconstruction of more historical landslide events should be carried out to test the capability of meshless methods applied to real cases.

*Author contributions.* All authors contribute to outline of the manuscript. LW and XZ programmed the code and performed the numerical
5  simulations. LW and ST wrote the manuscript with discussions and improvements with all authors.

*Competing interests.* The authors declare that they have no conflict of interest.

*Acknowledgements.* The author Liang Wang greatly appreciates the financial support from the cooperation agreement between the University of Bologna and the China Scholarship Council.



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
