# Peer review of "Shallow landslides modeling using a particle finite element model with emphasis on landslide evolution"

_Earth Surface Dynamics, 2019_

## Referee Comment (RC1) · Anonymous Referee #1 · 28 May 2019

**GENERAL COMMENTS**

The Authors present the numerical simulation of shallow landslides using a PFEM formulation already presented and published in other works of one of the Authors.

The PFEM is a widely used numerical strategy for the analysis of large-deformations problems. Originally, the method was proposed for free-surface fluid dynamics analysis (e.g. see works of S. Idelsohn, E. Oñate and co-workers), and then extended to other applications still involving large variations of topology of computational domains, such as in phase change problems, particulate-flow, and geotechnical applications, including also landslides simulation (e.g. see works of M. Cremonesi, F. Salazar, A. Larese, X.

Zhang and co-workers).

In this sense, the proposed method cannot be considered innovative either for the type of application (landslide simulation) or for the used numerical strategy (it bases on the work of Zhang, 2015).

From the analysis of the main case of the work, the Cà Mengoni landslide, it is hard to assess the accuracy of the numerical results. This is mainly due to the nature of the considered landslide event, whose large duration (around two days) and evolving external conditions are difficult to reproduce numerically. This explains the large differences between the numerical results and the observation, also after the calibration of the reduction factor. Furthermore, the Authors model a 3D problem with a 2D method without justifying/proving the validity of this approximation.

On the other hand, the first numerical test is quite simple and the deformations suffered from the computational domain are so small that the utility of the PFEM massive remeshing is questionable.

Apart from these important issues, the article lacks details of the used numerical method. In particular, the PFEM description is not clear and the method appears hard to understand for people not familiar with the method. Furthermore, some parts of the article are not clear or not properly justified, and the overall quality of pictures and graphs is low (see the following sections).

According to these considerations, although the method has surely a strong potential for landslides simulations (as already shown in other publications), I think that the paper cannot be published as it is. The lack of novelty of the work could be compensated by presenting the application (and validation) of the method against a real landslide event, preferably in three dimensions. In this sense, the second test cannot play this role for the reasons previously exposed. Furthermore, an extended revision of the work according to the following indications is suggested.

SPECIFIC COMMENTS

Section 1 – The authors should explain more clearly which are the novelties of the work and why the work is worth to be published.

Section 1 – The works of Salazar (2016) and Cremonesi (2011) on landslides simulation with the PFEM should be referenced.

Section 1 – Re-mapping operations are used in the PFEM only when historical variables are used and stored at nodes. This is not done in the classical (and most used) PFEM approach, which is generally used for fluid dynamics problem. Indeed, the PFEM for solid mechanics has been used only in a few works (J.M Carbonell, X. Zhang, W. Zhang and co-workers). This is not a weak point of the method at all, but it should be mentioned in the introduction.

Section 2 – The description of the PFEM is poor. It is not even mentioned how tessellation is built (I guess Delaunay Triangulation as in the standard PFEM). In this sense, Figure 2 is not clear and does not help to understand the method (the initial mesh is identical to the discretization after remeshing, so why remeshing?).

Section 2 – The explanation of remapping operations is not clear, and, again, Figure 2 does not help the understanding. Furthermore, from the picture, it seems that three integration points are used in the first mesh, and only one in the new one. Which elements are used? Linear or quadratic? This choice should be also motivated.

Section 2 – Some considerations about the effect of remeshing over stresses accuracy should be included.

Section 3 – Why are the Authors using different mesh sizes for Cases 1 and 2?

Section 3 – Why Case 1 is not compared to other numerical results? If a validation of Case 1 cannot be done, the test can be removed as it does not add any new information to the work.

Section 4 – All the section is a bit confusing and it is hard to follow and understand the motivation of the several tests done by the Authors. Furthermore, as already said previously, the accuracy of the numerical results cannot be clearly assessed.

TECHNICAL CORRECTIONS AND TYPOS

Section 1 - Please, define the anagrams SPH, MPM, PFEM and FEM

Section 2 – Please, find a more appropriate title for the section.

Section 2 – Please, rephrase the sentence 'PFEM is the particle extension of the classical FEM method' which is very vague.

Section 2 – Please, rephrase lines 11-13.

Section 3 - Please, do not make reference to Whang 2019 if the work has not already accepted.

Section 3 – Please, define the mesh size as a length and not as an area.

---

## Referee Comment (RC2) · Anonymous Referee #2 · 31 May 2019

The paper addresses the application of a PFEM to the study of landslides. The test cases are well known simple slope examples and a moderate-velocity landslide. Even though the numerical scheme is known and already applied for the study of some landslides, this application for a landslide of slower kinematic back analyzed on an adequate amount of monitoring data may be of interest for the readers.

However, there are some major issues that need to be addressed, I divide them in broad comments and specific comments.

BROAD COMMENTS

In the title and in the paper you refer to "shallow landslides". A landslide as Ca' Mengoni

that has a 30 m deep slip surface cannot be defined as shallow by any nomenclature standard. The term shallow should be discarded.

Some specifics about the difference between associated and non-associated and the flow rule should be inserted in the text. ESurf is a journal with broad audience and the basic assumption should be at least briefly stated. Moreover, the soil parameters, the location of the water table, the boundary conditions for all the landslide simulation should be added (maybe in a table). That would help understand better the results and also help ensure reproducibility.

Some performance index about the congruence of your results with the monitoring data like location of the slip surface and velocity of the landslide should also be present. The MI index landslide profile change cannot be the only parameter, in fact from the definition of landslide countermeasure works the proper representation of the location of the slip surface is in a sense more important.

The part about the "weakening process" is not well developed. I would have expected to see the function you use for simulating the weakening process rather than a somewhat vague description in paragraph 4.2.4. The function was calibrated just through back analysis? Was some rheological consideration included in the definition of the weakening function? You show that "it can be done" but your results does not provide a good representation of the process, so you have to explain better why you think that these results are interesting for the scientific community even if the work is still largely in progress.

SPECIFIC COMMENTS: l 21 p 1: rainfall does not affect "material strength", were you speaking of total stress?

Fig – 8: the figure is already published in another paper (consider copyright issues) – would it be better to modify it a little bit, maybe with the depth of the slip surface for each inclinometer

l 3 p 2: discard gradually

l 10 p 3: discard the before PFEM and FEM

l 15 p 3: mapping.

l 22 p 3: analyze dam stability

l 2 p 4: to prove that our model can be used to assess landslide hazard

l 7 p 8: discard deposited and add stopped/halted

Fig 9: can you please use the same scale for figure 9 as for figure 8? The whole shape of the slope in fig. 9 (and following) is distorted and it is not possible to compare the actual movement with the modelled one.

paragraph 4.2 l 7-11 p 10: The whole paragraph is not clear and needs to be properly re-written. Moreover, discard shallow depth

l 13-15 p 10: Correct from the adopted with using. Moreover, is it correct to say that you identified a slip surface when your FOS is way above 1.5? In fact then you say that you use a factor of 1.5 to reduce the parameters. So the whole first 4 lines of this paragraph need to be edited for clarity and following the proper order.

l 20 p 10: Berti placed the WT at ground level, where did you put it?

paragraph 4.2.1 so for the static analysis the non-associated model does not work properly – you need to discuss this result

paragraph 4.2.2: the 4 set of parameters used to produce the outputs of figure 11 should be summarized in a table, otherwise it is impossible to assess your results.

l15-16 p 11: are the virtual time and the actual time of failure somewhat comparable? what was the actual (target) maximum displacement?

l18 p 11: misfit index (MI)

Fig 12: check the scale distortion

[Figure]

---

## Author Comment (AC1) · 13 Jun 2019

We thank the Anonymous Referee #1 for his constructive comments. Liang Wang on behalf of all Co-Authors.

GENERAL COMMENTS:

(1) RC1: "The use of PFEM is not new in Landslide simulation"

AC: Agree. From the perspective of computational mechanics, PFEM is a widely used numerical strategy. However, in a geological approach, the efforts trying to link slope stability and landslide run-out analysis are of great interest to geotechnical and geological researchers. Recent studies can be found in e.g., Material Point Method (MPM) [1-2], Smoothed Particle Hydrodynamics (SPH) [3].

(2) RC1: "Model a 3D problem with a 2D method"

AC: The chosen case of the Cà Mengoni landslide was studied in a previous paper[4] by other authors with a 2D model to investigate stability analysis since most geotechnical data were acquired after the landslide occurrence only along a longitudinal transect. 2D models are indeed common practice even today[5]. So far, very few studies are conducted to cover the entire evolution of landslides, As for the 3D applications of PFEM to landslides [6-7], they are limited to the dynamic aspect (run-out analysis), which is usually done by using depth-averaged models.

(3) RC1: "Modeling a long duration landslide. . ."

AC: The model mainly presents the response of landslide geometry to the parameters in the framework of PFEM. The results indicate the existence of an intrinsic weakening process, which can provide the time evolution of the geo-material behavior. So far, we are not able to fully simulate this process since there are no observation data.

(4) RC1: "The first case is quite simple and the use of PFEM is questionable"

AC: The shallow failure mode has also been produced by other numerical methods, e.g. SPH[8] and MPM[2]. The whole computational configuration is similar to the experiment of the collapse of aluminum bars [9,10]. As for the question on remeshing and mapping, Figure 5 in the literature [10] (concerning the dam break problem) shows that the frequency of mapping doesn't play a significant role in such case. We will add a section to discuss the remeshing and mapping applied to the dynamic evolution of the slope.

(5) RC1 :". . .., an extended revision is suggested"

AC: Accepted. In the submitted version of our paper, we didn't explain PFEM in detail especially in the explanation of the PFEM for solid mechanics.

SPECIFIC COMMENTS

(1) RC1: "The authors should explain more clearly which are the novelties of the work and why the work is worth to be published."

AC: We investigated the application of PFEM to the whole process of the landslide motion from inception to the end and used a real case, for which few studies have been conducted. We will explain it better in revision.

(2) RC1: "The works of Salazar (2016) and Cremonesi (2011) on landslides simulation with the PFEM should be referenced."

AC: We will add them.

(3) RC1: "Re-mapping operations are used in the PFEM only when historical variables are used and stored at nodes. This is not done in the classical (and most used) PFEM approach, which is generally used for fluid dynamics problem. Indeed, the PFEM for solid mechanics has been used only in a few works (J.M Carbonell, X.Zhang, W. Zhang and co-workers). This is not a weak point of the method at all, but it should be mentioned in the introduction."

AC: We will explain better the use of the mapping technique in PFEM solid and its accuracy.

(4) RC1: "The description of the PFEM is poor. It is not even mentioned how tessellation is built (I guess Delaunay Triangulation as in the standard PFEM). In this sense, Figure 2 is not clear and does not help to understand the method (the initial mesh is identical to the discretization after remeshing, so why remeshing?)."

AC: A detailed procedure will be added.

(5) RC1: "The explanation of remapping operations is not clear, and, again, Figure 2 does not help the understanding. Furthermore, from the picture, it seems that three integration points are used in the first mesh, and only one in the new one. Which

elements are used? Linear or quadratic? This choice should be also motivated."

AC: We will explain the mixed triangular element we use and the procedure for mapping. We graphed only one new point to explain, but in fact mapping is done for all integration points. A clearer text will be put after figure.

(6) RC1: "Some considerations about the effect of remeshing over stresses accuracy should be included."

AC: The communication between historical variables, e.g. stress, at gauss points from old elements to new elements induces errors, which are however always minimized by reducing the time step. We will discuss it based on our present simulation.

(7)-(8) RC1: "The case 1..."

AC: Agree. Following the suggestion, we will remove Case 1.

(9) RC1: "All the section is a bit confusing and it is hard to follow and understand the motivation of the several tests done by the Authors. Furthermore, as already said previously, the accuracy of the numerical results cannot be clearly assessed."

AC: Our aim here was to investigate the difference between dynamic analysis and static analysis based on a real case in relatively simple conditions. The simulations are done to capture some dynamic process of the landslide based on the whole process simulation. We will improve the texts and figures in the revision. As for the accuracy, the whole computational configuration is similar to the experiment of the collapse of aluminum bars[ 9, 10].

TECHNICAL CORRECTIONS AND TYPOS:

Accept all and will be modified in the revision.

REFERENCES:

[1] Liu X, Wang Y, Li D Q. Investigation of slope failure mode evolution during large

deformation in spatially variable soils by random limit equilibrium and material point methods[J]. Computers and Geotechnics, 2019, 111: 301-312.

[2] Yerro A, Alonso E E, Pinyol N M. Run-out of landslides in brittle soils[J]. Computers and Geotechnics, 2016, 80: 427-439.

[3] Li L, Wang Y, Zhang L, et al. Evaluation of Critical Slip Surface in Limit Equilibrium Analysis of Slope Stability by Smoothed Particle Hydrodynamics[J]. International Journal of Geomechanics, 2019, 19(5): 04019032.

[4] Berti M, Bertello L, Bernardi A R, et al. Back analysis of a large landslide in a flysch rock mass[J]. Landslides, 2017, 14(6): 2041-2058.

[5] Yerro A, Soga K, Bray J. Runout evaluation of Oso landslide with the material point method[J]. Canadian Geotechnical Journal, 2018 (999): 1-14.

[6] Cremonesi M, Ferri F, Perego U. A basal slip model for Lagrangian finite element simulations of 3D landslides[J]. International Journal for Numerical and Analytical Methods in Geomechanics, 2017, 41(1): 30-53.

[7] Franci A, Cremonesi M. 3D regularized $\mu$ (I)-rheology for granular flows simulation[J]. Journal of Computational Physics, 2019, 378: 257-277.

[8] Peng C, Wu W, Yu H, et al. A SPH approach for large deformation analysis with hypoplastic constitutive model[J]. Acta Geotechnica, 2015, 10(6): 703-717.

[9] Bui H H, Fukagawa R, Sako K, et al. Lagrangian meshfree particles method (SPH) for large deformation and failure flows of geomaterial using elastic–plastic soil constitutive model[J]. International Journal for Numerical and Analytical Methods in Geomechanics, 2008, 32(12): 1537-1570.

[10] Zhang X, Oñate E, Torres S A G, et al. A unified Lagrangian formulation for solid and fluid dynamics and its possibility for modelling submarine landslides and their consequences[J]. Computer Methods in Applied Mechanics and Engineering, 2019, 343:

314-338.

---

## Author Comment (AC2) · 13 Jun 2019

We wish to thank the Anonymous Referee #2 for his/her insightful and helpful suggestions. Liang Wang on behalf of all Co-Authors.

A) GENERAL COMMENTS:

(1) RC2: "The term shallow should be discarded".

AC: In the paper, we used the adjective "shallow" to emphasize the small-depth movements from the perspective of continuum modeling. However we accept that the term "shallow landslide" is not suitable as regards landslide classification, and we will correct

it in the revision.

(2) RC2: "Some specifics about the difference between associated and non-associated and the flow rule should be inserted in the text. Moreover, the soil parameters, the location of the water table, the boundary conditions for all the landslide simulation should be added (maybe in a table)."

AC: Accepted. "We will add texts to explain these differences and the details of simulations will be put into a table."

(3) RC2: "Some performance index about the congruence of your results with the monitoring data like location of the slip surface and velocity of the landslide should also be present. The MI index landslide profile change cannot be the only parameter, in fact from the definition of landslide countermeasure works the proper representation of the location of the slip surface is in a sense more important".

AC: Accepted. "We will add a section to address the dynamic evolution of the landslide."

(4) RC2: "The part about the 'weakening process' is not well developed. I would have expected to see the function you use for simulating the weakening process rather than a somewhat vague description in paragraph 4.2.4. The function was calibrated just through back analysis? Was some rheological consideration included in the definition of the weakening function? You show that "it can be done" but your results do not provide a good representation of the process, so you have to explain better why you think that these results are interesting for the scientific community even if the work is still largely in progress."

AC: We added a time function to gradually decrease strength parameters, i.e. cohesion and internal friction angle, since the present model handles the slope geometry response to material parameters. However, we do not know the real material weakening information, so the time-function is used in the paper to distinguish between a dynamic-process and a quasi-static process. The 'weakening process' is important

since it links the initiation and the propagation stages of the landslide. Further it has been shown that the run-out distance of the landslide is short without a 'weakening process'. Similar efforts trying to bridge the classical slope stability analysis and the run-out analysis by simulating the whole process of landslide can be seen in recently published works, e.g. MPM (material point methods) [1] or SPH (Smoothed Particle Hydrodynamics) [3]. In the revision paper, we will try to implement a strain-soften model (e.g. [2], [4], [5]), to capture the dynamic process and compare with the existing results.

B) SPECIFIC COMMENTS:

(1) RC2: rainfall does not affect "material strength", were you speaking of total stress?

AC: Accept and will be modified.

(2) RC2: "Modify the Figure 8"

AC: We will ask the original figure and modify it.

(3) RC2: "discard gradually"

AC: Accepted.

(4) RC2:"discard the before PFEM and FEM"

AC: Accepted.

(5) RC2: "Mapping"

AC: Accepted.

(6) RC2: "Analyze dam stability"

AC: Accepted.

(7) RC2: "to prove that our model can be used to assess landslide hazard"

AC: we will explain it better

(8) RC2: "discard deposited and add stopped/halted"

AC: Accepted.

(9) RC2: "use the same scale for Figure 9 and Figure 8"

AC: It will be done.

(10) RC2: "paragraph 4.2 l 7-11 p 10: The whole paragraph is not clear"

AC: It will be explained better.

(11) RC2: "13-15 p 10: Correct from the adopted with using...; the whole first 4 lines of this paragraph need to be edited for clarity and following the proper order"

AC: we will explain it better.

(12) RC2: "Berti placed the WT at ground level, where did you put it?"

AC: The hydrological condition is not clear for this case, Prof. Berti placed different ground level to back-analyze the slip-surface. In our model, we do not consider the hydrological condition.

(13) RC2: "paragraph 4.2.1 so for the static analysis the non-associated model does not work properly you need to discuss this result"

AC: There is a large difference between the associated and non-associated model for this case. Two different failures, i.e. local failure and a clear slip surface, are observed. This can be attributed to the slope geometry and material parameters. We also present in Figure11 (a) that the slip surface defined by static analysis is not consistent with the one resulting from dynamic analysis. This also indicates that for some complex landslides, the dynamic analysis might be more appropriate.

(14) RC2: "the 4 set of parameters used to produce the outputs of figure 11 should be summarized in a table, otherwise it is impossible to assess your results."

AC: We will do it.

(15) RC2: "l15-16 p 11: are the virtual time and the actual time of failure somewhat comparable? what was the actual (target) maximum displacement?"

AC: The virtual time and actual time can be comparable if we define a similar time function to reduce the material parameter. The actual maximum displacement is not investigated by surveys, while the elevation change was calculated by pre-failure and post-failure profile.

(16) RC2: "l18 p 11: misfit index (MI)"

AC: Accepted.

REFERENCES:

[1] Liu X, Wang Y, Li D Q. Investigation of slope failure mode evolution during large deformation in spatially variable soils by random limit equilibrium and material point methods[J]. Computers and Geotechnics, 2019, 111: 301-312.

[2] Yerro A, Soga K, Bray J. Runout evaluation of Oso landslide with the material point method[J]. Canadian Geotechnical Journal, 2018 (999): 1-14.

[3] Li L, Wang Y, Zhang L, et al. Evaluation of Critical Slip Surface in Limit Equilibrium Analysis of Slope Stability by Smoothed Particle Hydrodynamics[J]. International Journal of Geomechanics, 2019, 19(5): 04019032.

[4] Zhang X, Sloan S W, Oñate E. Dynamic modelling of retrogressive landslides with emphasis on the role of clay sensitivity[J]. International Journal for Numerical and Analytical Methods in Geomechanics, 2018, 42(15): 1806-1822.

[5] Troncone A. Numerical analysis of a landslide in soils with strain-softening behaviour[J]. Geotechnique, 2005, 55(8): 585-596.

---

## Editor Comment (EC1) · Xuanmei Fan (Editor) · 23 Jun 2019

Thank you for submitting your work to Earth Surface Dynamics. After checking the reviewers' comments and the paper carefully, regretfully, I cannot encourage you to submit the revision. Both reviewers questioned about the innoviation of the paper and also the calibration of the model (which seems too much simplified). Personally I think the paper fits better to journals like "Landslides" or "Engineering Geology", if the authors can really address what is new of the model and how it can be applied to reproduce landslide dynamics.

[Figure]

2019.